# Vps8 overexpression inhibits HOPS-dependent trafficking routes by outcompeting Vps41/Lt

Péter Lőrincz[1,2]*, Lili Anna Kenéz[1], Sarolta Tóth[1], Viktória Kiss[3], Ágnes Varga[1], Tamás Csizmadia[1], Zsófia Simon-Vecsei[1], Gábor Juhász[1,3]*

[1]Department of Anatomy, Cell and Developmental Biology, Eötvös Loránd University, Budapest, Hungary; [2]Premium Postdoctoral Research Program, Hungarian Academy of Sciences, Budapest, Hungary; [3]Institute of Genetics, Biological Research Centre, Hungarian Academy of Sciences, Szeged, Hungary

**Abstract** Two related multisubunit tethering complexes promote endolysosomal trafficking in all eukaryotes: Rab5-binding CORVET that was suggested to transform into Rab7-binding HOPS. We have previously identified miniCORVET, containing Drosophila Vps8 and three shared core proteins, which are required for endosome maturation upstream of HOPS in highly endocytic cells (Lőrincz et al., 2016a). Here, we show that Vps8 overexpression inhibits HOPS-dependent trafficking routes including late endosome maturation, autophagosome-lysosome fusion, crinophagy and lysosome-related organelle formation. Mechanistically, Vps8 overexpression abolishes the late endosomal localization of HOPS-specific Vps41/Lt and prevents HOPS assembly. Proper ratio of Vps8 to Vps41 is thus critical because Vps8 negatively regulates HOPS by outcompeting Vps41. Endosomal recruitment of miniCORVET- or HOPS-specific subunits requires proper complex assembly, and Vps8/miniCORVET is dispensable for autophagy, crinophagy and lysosomal biogenesis. These data together indicate the recruitment of these complexes to target membranes independent of each other in Drosophila, rather than their transformation during vesicle maturation.

DOI: https://doi.org/10.7554/eLife.45631.001

*For correspondence:
concrete05@gmail.com (PL);
szmrt@elte.hu (GJ)

**Competing interests:** The authors declare that no competing interests exist.

## Introduction

Lysosomal degradation is essential for the survival and homeostasis of eukaryotic cells. The two main routes of lysosomal degradation are endocytosis and autophagy, and HOPS (homotypic fusion and vacuole protein sorting) tethering complex is a central player in both processes. HOPS was identified in yeast and is defined by two Ypt7 (Rab7 in higher eukaryotes) binding subunits Vps41 and Vps39 on its opposing ends (*Balderhaar et al., 2013*; *Balderhaar and Ungermann, 2013*; *Bröcker et al., 2012*; *Lürick et al., 2018*; *Peplowska et al., 2007*; *Plemel et al., 2011*; *Rieder and Emr, 1997*; *Seals et al., 2000*; *Wickner and Schekman, 2008*; *Wurmser et al., 2000*). In metazoan cells including Drosophila, HOPS directly binds to Rab2 and Rab7-binding adaptors to ensure fusions of lysosomes with autophagosomes, late endosomes, secretory granules and Golgi derived vesicles (*Angers and Merz, 2009*; *Balderhaar and Ungermann, 2013*; *Csizmadia et al., 2018*; *Fujita et al., 2017*; *Jiang et al., 2014*; *Kajiho et al., 2016*; *Lőrincz et al., 2017b*; *McEwan et al., 2015*; *Pankiv et al., 2010*; *Pols et al., 2013a*; *Solinger and Spang, 2013*; *van der Kant et al., 2013*; *Wang et al., 2016*).

A closely related multisubunit complex termed CORVET (Class C core endosome vacuole tethering) mediates the tethering and fusions of Vps21 (Rab5 in higher eukaryotes) positive membranes (*Balderhaar et al., 2013*; *Balderhaar and Ungermann, 2013*; *Bröcker et al., 2012*; *Lürick et al.,*

*2018*; *Peplowska et al., 2007*; *Plemel et al., 2011*; *Rieder and Emr, 1997*; *Seals et al., 2000*; *Wurmser et al., 2000*). Both CORVET and HOPS share a common core of class C Vps proteins (Vps11, Vps16, Vps18 and Vps33), but in the former complex two Vps21 (Rab5 in higher eukaryotes) binding subunits: Vps8 and Vps3 are present instead of the Ypt7/Rab7 binding Vps41 and Vps39, respectively (*Balderhaar and Ungermann, 2013*; *Bröcker et al., 2012*; *Lürick et al., 2018*). Whilst HOPS is conserved across metazoans, higher eukaryotes lack Vps3, which is therefore yeast-specific. Mammalian CORVET contains Vps39-2 (also known as Tgfbrap1 or Trap1) in the place of Vps3, and Vps8 is conserved (*Lachmann et al., 2014*; *Perini et al., 2014*). Drosophila has a smaller CORVET variant termed miniCORVET, containing Vps8 and only three of the four class C Vps proteins (Dor/Vps18, Car/Vps33A and Vps16A). Thus, Vps11 is a HOPS specific protein in flies (*Lőrincz et al., 2016a*).

Although CORVET and HOPS complexes share common subunits, the question whether these complexes assemble de novo or they can be converted into each other is still open. In yeast, a series of biochemical experiments on overexpressed complex specific subunits suggested the existence of intermediate complexes that contain one CORVET and one HOPS specific Vps protein (*Peplowska et al., 2007*). Moreover, overexpression of CORVET specific subunits can disturb endosome maturation and Vps3 can displace Vps39 from HOPS, potentially as a result of competition between complex specific subunits (*Markgraf et al., 2009*; *Ostrowicz et al., 2010*; *Peplowska et al., 2007*). These results raise the possibility that during assembly, complex-specific proteins may compete for class C proteins in yeast.

Others and we have previously shown that Drosophila is an excellent model to study miniCORVET and HOPS mediated vesicular trafficking processes, including endosome maturation in nephrocytes, autophagosome-lysosome fusion in fat cells, crinophagy in salivary glands and eye pigment granule biogenesis (*Akbar et al., 2009*; *Csizmadia et al., 2018*; *Lindmo et al., 2006*; *Lőrincz et al., 2016a*; *Lőrincz et al., 2017a*; *Lőrincz et al., 2016b*; *Lőrincz et al., 2017b*; *Pulipparacharuvil et al., 2005*; *Sevrioukov et al., 1999*; *Takáts et al., 2014*; *Takáts et al., 2015*; *Warner et al., 1998*). We now aimed to answer the question whether the overexpression of CORVET-specific Vps8 or its HOPS-specific counterpart Vps41 could affect HOPS or CORVET dependent processes, respectively, and if so how.

Through a series of confocal and electron microscopy experiments, we show that overexpression of Vps8 inhibits HOPS dependent trafficking, such as late endosome maturation in nephrocytes, autophagosome-lysosome fusion in fat cells, crinophagy in salivary glands and pigment granule biogenesis in eyes. We also found that similar to the loss of HOPS, class C Vps core proteins or selected small GTPases, the late endosomal localization of Vps41 is lost in Vps8 overexpressing cells. Based on co-immunoprecipitation data, we show that the amount of HOPS decreases in Vps8 overexpressing animals, suggesting that Vps8 may negatively regulate HOPS by outcompeting Vps41. Since yeast Vps8 was suggested to be involved in autophagosome formation (*Chen et al., 2014*; *Zhou et al., 2017*) the possible function of miniCORVET was also examined, but we found that this feature of CORVET is not conserved.

## Results and discussion

### Overexpression of Vps8 results in a HOPS mutant-like phenotype in garland nephrocytes, while the overexpression of Vps41-9xHA has no effect

Garland nephrocytes in Drosophila are excellent model cells to study endosomal traffic because the loss of endosomal tethering proteins leads to a very specific change in the appearance of the endosomal compartments. Briefly, the loss of proteins required for early endosomal fusions (such as Vps8, Vps16a, Vps18, Vps33a and Rabenosyn-5) leads to the fragmentation of Rab7 positive late endosomes, while the loss of late endosomal-lysosomal tether HOPS (Vps11, Vps41, Vps39) leads to the enlargement of Rab7 positive endosomes (*Lőrincz et al., 2016a*). Due to the accumulation of late endosomes, these cells are also enlarged compared to control cells (*Lőrincz et al., 2016a*).

As Vps8 and Vps41 are suggested to occupy the same binding site in CORVET and HOPS, respectively (*Peplowska et al., 2007*), we tested whether the overexpression of a complex specific subunit may result in the inhibition of the other complex. In support of this model, yeast data suggested that competition may exist between the CORVET and HOPS complex specific subunits (*Markgraf et al., 2009*; *Ostrowicz et al., 2010*; *Peplowska et al., 2007*). To examine this possibility in metazoan cells, we overexpressed either the miniCORVET specific Vps8 or the HOPS specific Vps41 in garland nephrocytes. Strikingly, we found that – similar to *vps11* or *vps39* RNAi cells - Vps8 overproducing cells are swollen (even in a Vps8 mutant background) and contain enlarged late endosomes based on both confocal and electron microscopy (*Figure 1A–D,F–I*, *Figure 1—figure supplement 1*). Importantly this phenotype resembles to the yeast data: Vps8 overexpression delayed late endosome maturation and caused the accumulation of multivesicular bodies (MVBs) proximal to the vacuole (*Markgraf et al., 2009*) and the overproduction of Vps3 (yeast specific CORVET subunit) caused vacuole fragmentation (*Peplowska et al., 2007*), which is a definitive characteristic of HOPS loss-of function in yeast cells (*Raymond et al., 1992*). These data suggest that in both metazoan and yeast cells, the overproduction of CORVET-specific subunits may interfere with HOPS function. There are also important differences between yeast and fly cells. First, Vps8 overexpression in Drosophila completely phenocopied HOPS loss-of-function and apparently blocked late endosome to lysosome fusion, unlike only causing a delay of endosome maturation observed in yeast. Second, Vps8 overproduction caused punctate accumulation of Vps21 (Rab5 equivalent in yeast) on membranes in a Vps3-dependent fashion (*Peplowska et al., 2007*) and MVBs in Vps8 overexpressing cells were also Vps21 positive (*Markgraf et al., 2009*). These suggest that CORVET specific Vps proteins promote Vps21 localization in yeast cells. In contrast, Rab5 positive endosomes kept their peripheral localization in Vps8 overexpressing fly cells just like in HOPS depleted cells (*Figure 1A–D*), suggesting that in higher eukaryotes, (mini)CORVET is dispensable for the targeting and stability of Rab5 on endosomal membranes.

Based on these Vps8 overexpression phenotypes, one might wonder whether Vps41 overexpression results in a miniCORVET loss-of-function phenotype. To our surprise this was not the case, as Vps41-9xHA overexpression had no effect on garland cell or late endosome size (*Figure 1A,E*, *Figure 1—figure supplement 1A,E–G*), changes in which are evident during Vps8 loss-of-function. To exclude the possibility of no transgene expression or the expression of a non-functioning protein (caused by the addition of the HA-tag), we overexpressed Vps41-9xHA in the nephrocytes of Vps41 mutant animals and visualized Rab7 and the HA-tag by immunostaining the cells. Control and Vps41 mutant cells lacked HA signal as expected, and the size of both the cells and Rab7-positive late endosomes were strikingly enlarged in the case of Vps41 mutants (*Figure 1—figure supplement 2A,B,D*). Importantly, Vps41-9xHA expression completely rescued the Vps41 mutant phenotype: the size of both the cells and their late endosomes was similar to controls. Moreover, the Vps41-9xHA signal could be detected on Rab7-positive endosomes, together indicating that functional protein is expressed from the *vps41-9xHA* transgene (*Figure 1—figure supplement 2C*). In line with this, GFP-Vps41 localized to late endosomes and vacuole contact sites in yeast, and the overexpression of Vps41 did not cause obvious defects, aside from a mild impairment of the AP-3 pathway (*Cabrera et al., 2010*; *Cabrera et al., 2009*). These together suggest that if competition occurs between the complex specific subunits, then (mini)CORVET specific one(s) may have higher affinity towards class C Vps proteins than HOPS specific Vps protein(s) do in metazoan cells.

## Overexpression of Vps8 inhibits autophagosome clearance

In addition to endocytosis, the other main lysosomal degradation route is autophagy. During the main pathway, autophagosomes isolate a portion of the cytoplasm and then fuse with lysosomes to degrade the sequestered cellular components. The larval fat tissue of Drosophila is a widely used model to study this process, and several established tools are available to monitor autophagy in flies (*Lőrincz et al., 2017a*; *Mauvezin et al., 2014*; *Nagy et al., 2015*).

The larval fat tissue of intensively growing (feeding) larvae is practically devoid of acidic lysosomes, while starvation induces numerous bright lysotracker positive autolysosomes during an autophagy response (*Lőrincz et al., 2017a*; *Scott et al., 2004*). We found that the number of lysotracker positive lysosomes dramatically decreased in Vps8 overexpressing clone cells (marked by the co-expression of GFP) compared to surrounding control cells in mosaic animals, indicating the impairment of starvation-induced autophagy (*Figure 2A*).

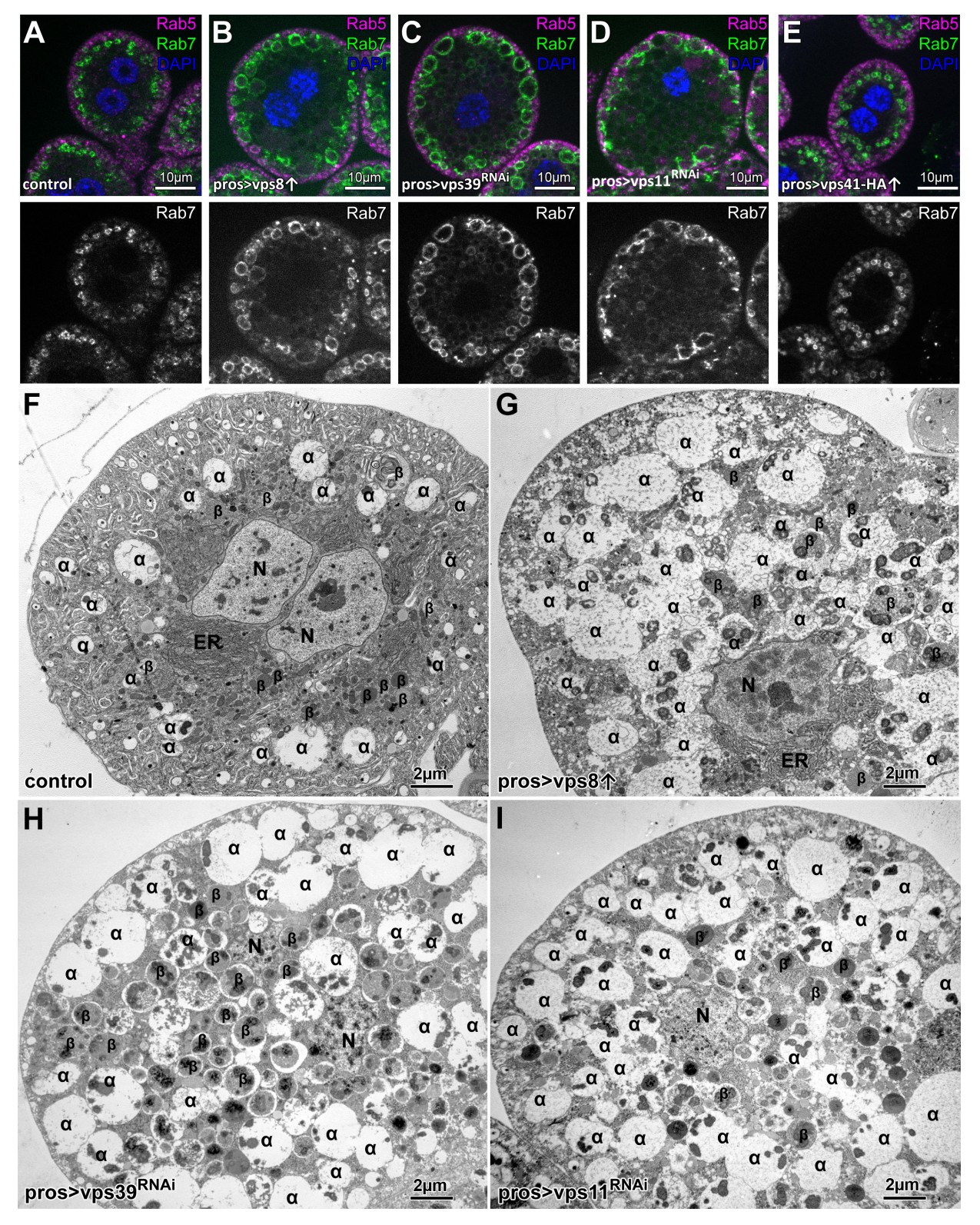

**Figure 1.** Overexpression of Vps8, but not Vps41 impairs endosome–lysosome fusion in *Drosophila* nephrocytes. (A–E) Rab7+ late, but not Rab5+ early endosomes are enlarged in Vps8 overexpressing garland nephrocytes (B) similar to *vps39* (C) or *vps11* (D) RNAi cells. The size and location of endosomes are similar in control (A) and Vps41-9xHA overexpressing cells (E). (F–I) Ultrastructural analyses of nephrocytes reveal that compared to
*Figure 1 continued on next page*

*Figure 1 continued*

controls (**F**), late endosomes (also known as α-vacuoles – indicated by α in the panels) are enlarged and contain multiple dense cores in Vps8 overexpressing cells (**G**), similar to *vps39* (**H**) or *vps11* (**I**) RNAi cells. β: β-vacuoles/lysosomes, ER: endoplasmic reticulum, N: nucleus.

DOI: https://doi.org/10.7554/eLife.45631.002

The following figure supplements are available for figure 1:

**Figure supplement 1.** additional nephrocyte data and the quantification of fluorescent data.

DOI: https://doi.org/10.7554/eLife.45631.003

**Figure supplement 2.** Overexpression of Vps41-9xHA rescues the *vps41* mutant phenotype and the protein localizes to late endosomes.

DOI: https://doi.org/10.7554/eLife.45631.004

Similar to our nephrocyte data, these suggested that Vps8 overexpression inhibits autophagosome clearance, a process that requires the HOPS tethering complex (*McEwan et al., 2015*; *Rieder and Emr, 1997*; *Takáts et al., 2014*). This was further supported by starvation experiments using the 3xmCherry-Atg8a reporter that labels both autophagosomes and autolysosomes because mCherry retains its fluorescence and accumulates inside the acidic milieu of lysosomes (*Lőrincz et al., 2017a*). This reporter revealed that similar to *vps39* RNAi cells, Vps8 overexpressing cells are indeed almost devoid of bright autolysosomes, only 'clouds' of many faint autophagosomes could be detected around the nuclei (*Figure 2B,C*). This phenotype also suggests autophagosome accumulation, which was confirmed using endogenous Atg8a immunostaining. Since endogenous Atg8a is degraded in the lysosomes unlike the 3xmCherry-Atg8a reporter, this staining labels mainly autophagosomes (*Lőrincz et al., 2017a*). Again similar to starved *vps39* RNAi cells, Vps8 overproducing cells contained an elevated number of Atg8a dots (*Figure 2D,E*), indicating a defect in autophagic flux.

Since yeast Vps8 was suggested to be involved in autophagy (*Chen et al., 2014*; *Zhou et al., 2017*), we used two independent RNAi lines to knock down Vps8 and determine whether such a role is conserved in higher eukaryotes. We could find no difference in the pattern of 3xmCherry-Atg8a or endogenous Atg8a signal between starved control and RNAi cells, indicating that in higher eukaryotes Vps8 - CORVET may not be involved in autophagy (*Figure 2—figure supplement 1*). Importantly, expressing the same RNAi constructs in garland nephrocytes of genomic promoter-driven Vps8-9xHA expressing animals abolished the HA signal in these cells and phenocopied the loss of Vps8 as these cells were swollen and filled with small fragmented late endosomes in case of both RNAi lines (*Figure 2—figure supplement 2*).

We utilized null mutants to further support our RNAi data that Vps8/miniCORVET is dispensable for autophagy using nephrocytes, a cell type in which Vps8 loss has a dramatic effect on endosome maturation (*Lőrincz et al., 2016a*). We visualized Atg8a-positive autophagosomes and p62/Ref(2)P, an autophagic receptor for poly-ubiquitinated proteins that forms protein aggregates and accumulates upon inhibition of autophagy. HOPS-specific subunit mutants for Vps41/Lt and Vps11 as well as the class C Vps16A caused an obvious increase in the number of both Atg8a and p62 dots, indicating a block of autophagosome turnover (*Figure 2—figure supplement 3A,C–E*). At the same time, Vps8 null mutants were indistinguishable from controls regarding punctate Atg8a and p62 signals (*Figure 2—figure supplement 3A,B*).

We then returned to Vps8 gain-of-function experiments and analyzed autophagic flux, that is, the turnover of material to be degraded. Organism-wide overexpression of Vps8 led to an obvious upregulation of p62 and both forms of Atg8a in well-fed adult flies, indicating a block of autophagic turnover (*Figure 2F*).

The mCherry-GFP-Atg8a reporter is commonly used to monitor autophagic flux because GFP signal is quenched in acidic lysosomes while mCherry persists (*Lőrincz et al., 2017a*; *Mauvezin et al., 2014*; *Nagy et al., 2015*). Compared to controls, the overexpression of Vps8 prevented the quenching of GFP: small dots double positive for both GFP and mCherry accumulated in starved fat cells (*Figure 3A,B*, *Figure 3—figure supplement 1*), indicating defective autophagic flux. These phenotypes are hallmarks of HOPS loss-of-function and are caused by the inability of autophagosomes to fuse with lysosomes in the absence of this tether (*Takáts et al., 2014*). We also looked at the colocalization of 3xmCherry-Atg8a with a lysosomal membrane protein fused to GFP (GFP-Lamp1) in starved fat cells. In controls, large 3xmCherry dots are positive for GFP-Lamp1 indicating ongoing fusions of autophagosomes with lysosomes, but the overlap of these signals dramatically decreased

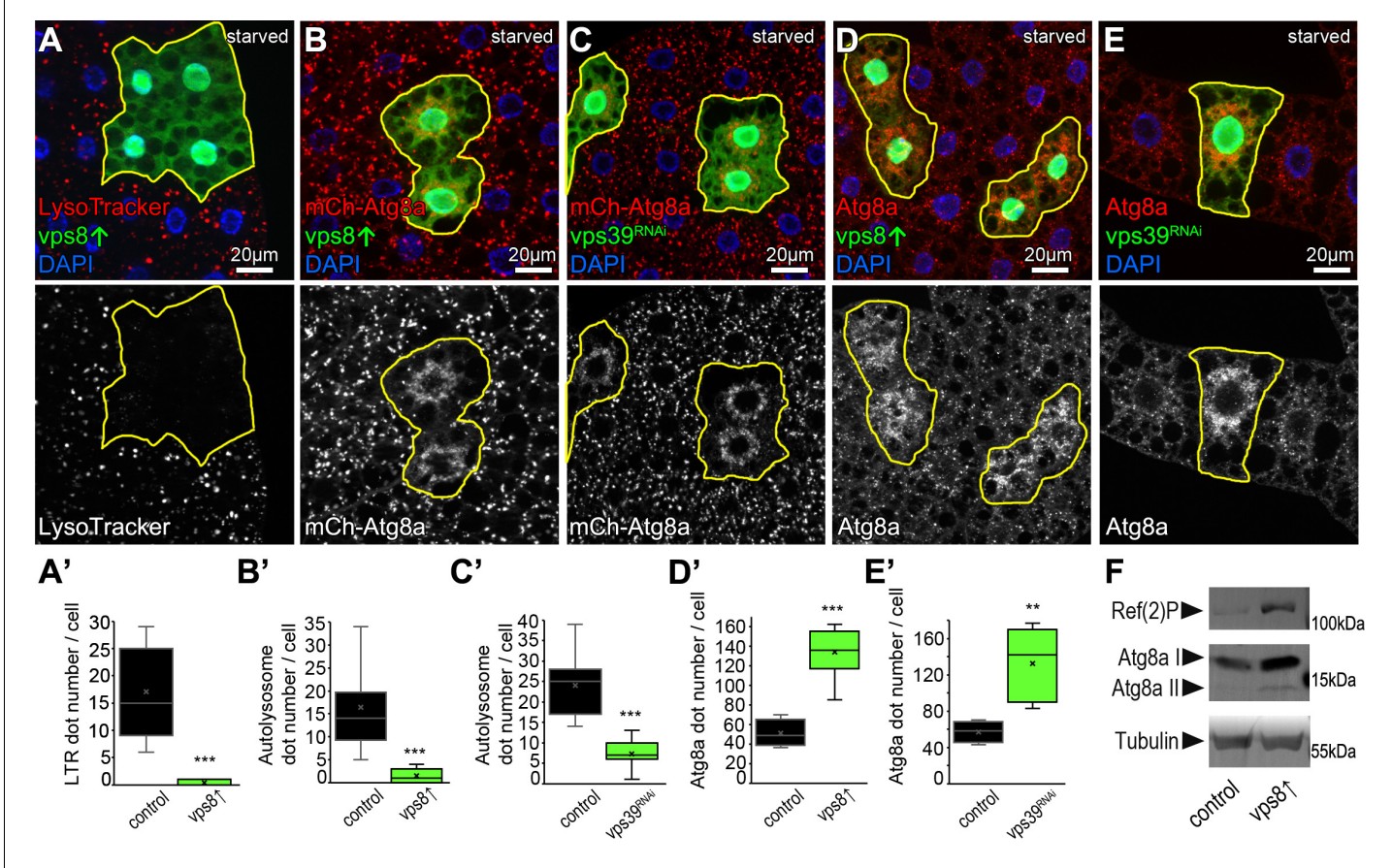

**Figure 2.** Overexpression of Vps8 inhibits autolysosome formation in starved fat cells, similar to HOPS (vps39) RNAi. (A) Overexpression of Vps8 in GFP + fat cells impairs LysoTracker Red dot formation compared with neighboring non-GFP control cells, similarly to HOPS loss-of-function (*Takáts et al., 2014*). (B and C) Both Vps8 overexpression (B) and *vps39* RNAi (C) impairs the proper formation of 3xmCherry-Atg8a+ autophagic vesicles in GFP + cells: these red dots are bigger and brighter in surrounding control cells and GFP+ cells contain smaller and fainter dots (likely autophagosomes) in both cases. (D and E) The number of endogenous Atg8a puncta (autophagosomes) increases in GFP+ Vps8 overexpressing (D) or *vps39* RNAi (E) cells compared to GFP-negative control cells. (A'–E') Quantification of data from panels A–E. The median and the average are indicated as a horizontal black line and x within the boxes, respectively. Bars show the upper and lower quartiles, and significant differences are indicated. (F) Western blot from well-fed adult lysates shows the obvious accumulation of Ref(2)P/p62 and both unlipidated (I) and autophagosome-associated, lipidated (II) forms of Atg8a in animals systemically overexpressing Vps8.

DOI: https://doi.org/10.7554/eLife.45631.005

The following figure supplements are available for figure 2:

**Figure supplement 1.** Vps8 (miniCORVET) is dispensable for autophagy in starved fat cells.
DOI: https://doi.org/10.7554/eLife.45631.006

**Figure supplement 2.** Validation of knockdown efficiencies for Vps8 RNAi lines used in this study.
DOI: https://doi.org/10.7554/eLife.45631.007

**Figure supplement 3.** Analysis of basal autophagy in nephrocytes.
DOI: https://doi.org/10.7554/eLife.45631.008

upon Vps8 overexpression (*Figure 3C,D*, *Figure 3—figure supplement 1*). These suggest the impairment of autophagosome-lysosome fusion in Vps8 overexpressing cells, causing autophagosomes and small lysosomes to accumulate.

Accordingly, the size of the lysosomal marker dLamp-3xmCherry positive dots decreased and their number increased in starved Vps8 overproducing cells, similar to Rab7-positive structures that represent both autophagosomes and lysosomes in these cells (*Hegedűs et al., 2016*) (*Figure 4A,C*, *Figure 4—figure supplement 1*). Both phenotypes resembled HOPS-specific *vps39* RNAi cells, further supporting that Vps8 overexpression leads to near-complete HOPS inhibition (*Figure 4B,D*,

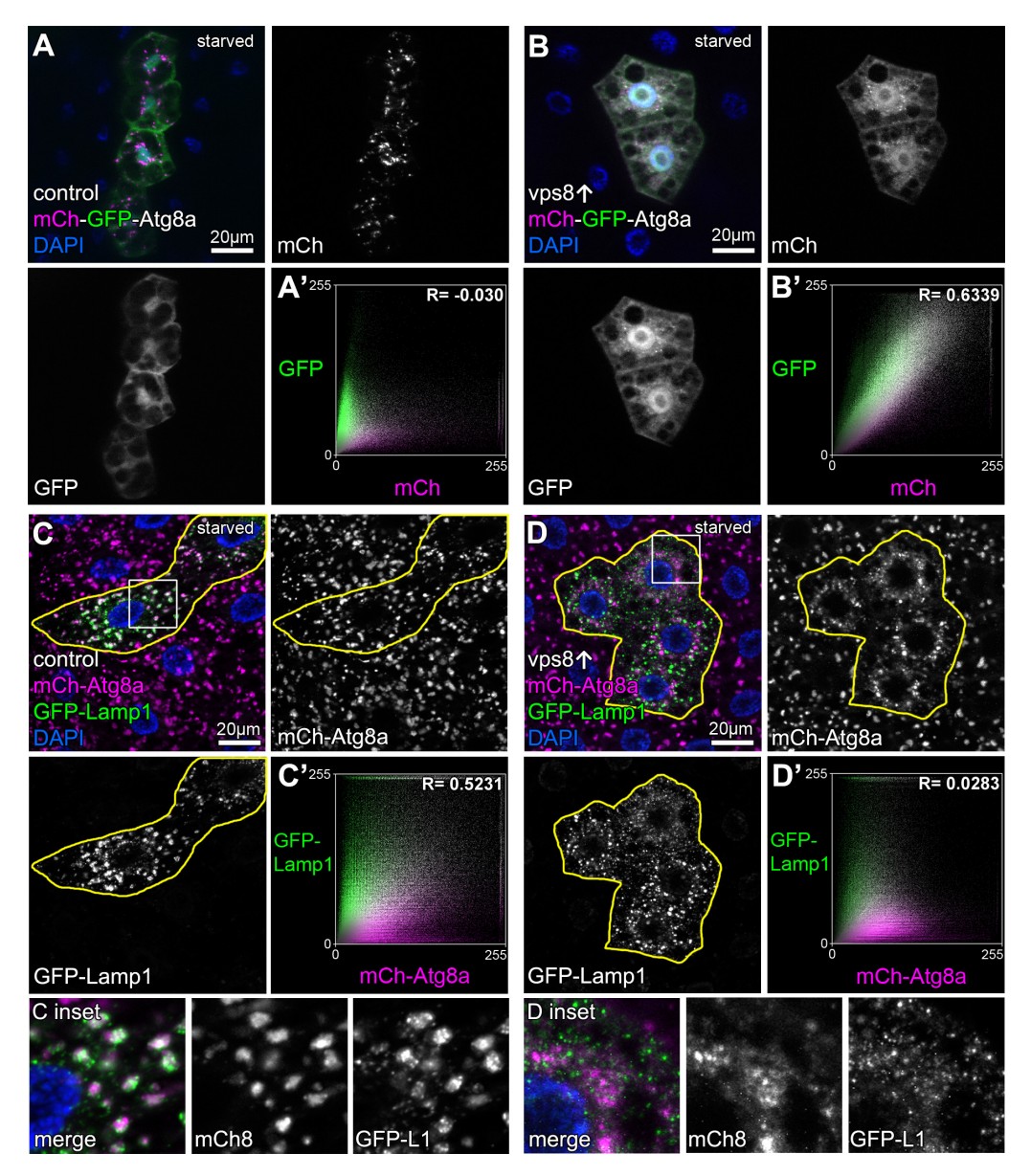

**Figure 3.** Overexpression of Vps8 inhibits autophagosome clearance in starved fat cells. Tandem mCherry-GFP-Atg8a experiments demonstrates autophagic flux in starved control cells, as GFP is quenched in lysosomes while punctate mCherry signal is retained (**A**). In contrast, GFP also remains fluorescent and colocalizes with mCherry in Vps8 overexpressing cells (**B**). (**A' and B'**) Averaged scatter plots (generated from 16 control and 13 UAS-Vps8 cells) show the intensity correlation profile of GFP with mCherry. Pearson correlation coefficients shown at the top of the panels indicate that in controls (**A'**) there is hardly any colocalization due to the autolysosomal quenching of GFP, whilst its signal persists in Vps8 overexpressing cells (**B'**), indicating the impairment of autophagic flux. (**C and D**) Large GFP-Lamp1 and 3xmCherry-Atg8a dots overlap in starved control cells (**C**) indicating proper autolysosome formation, while there is dramatically decreased overlap in Vps8 overexpressing cells (**D**). Note that insets show the boxed regions enlarged from C and D, respectively. (**C' and D'**) Averaged scatter plots (generated from 16 control and 20 UAS-Vps8 cells) show the intensity correlation profile of GFP-Lamp1 with 3xmCherry-Atg8a. Pearson correlation coefficients shown at the top of the panels indicate that the two signals colocalize in controls (**C'**), unlike in Vps8 overexpressing cells (**D'**), pointing to the impairment of autophagosome-lysosome fusion.
DOI: https://doi.org/10.7554/eLife.45631.009

The following figure supplement is available for figure 3:

**Figure supplement 1.** Quantification of data from *Figure 3*, panels A'-D'.
DOI: https://doi.org/10.7554/eLife.45631.010

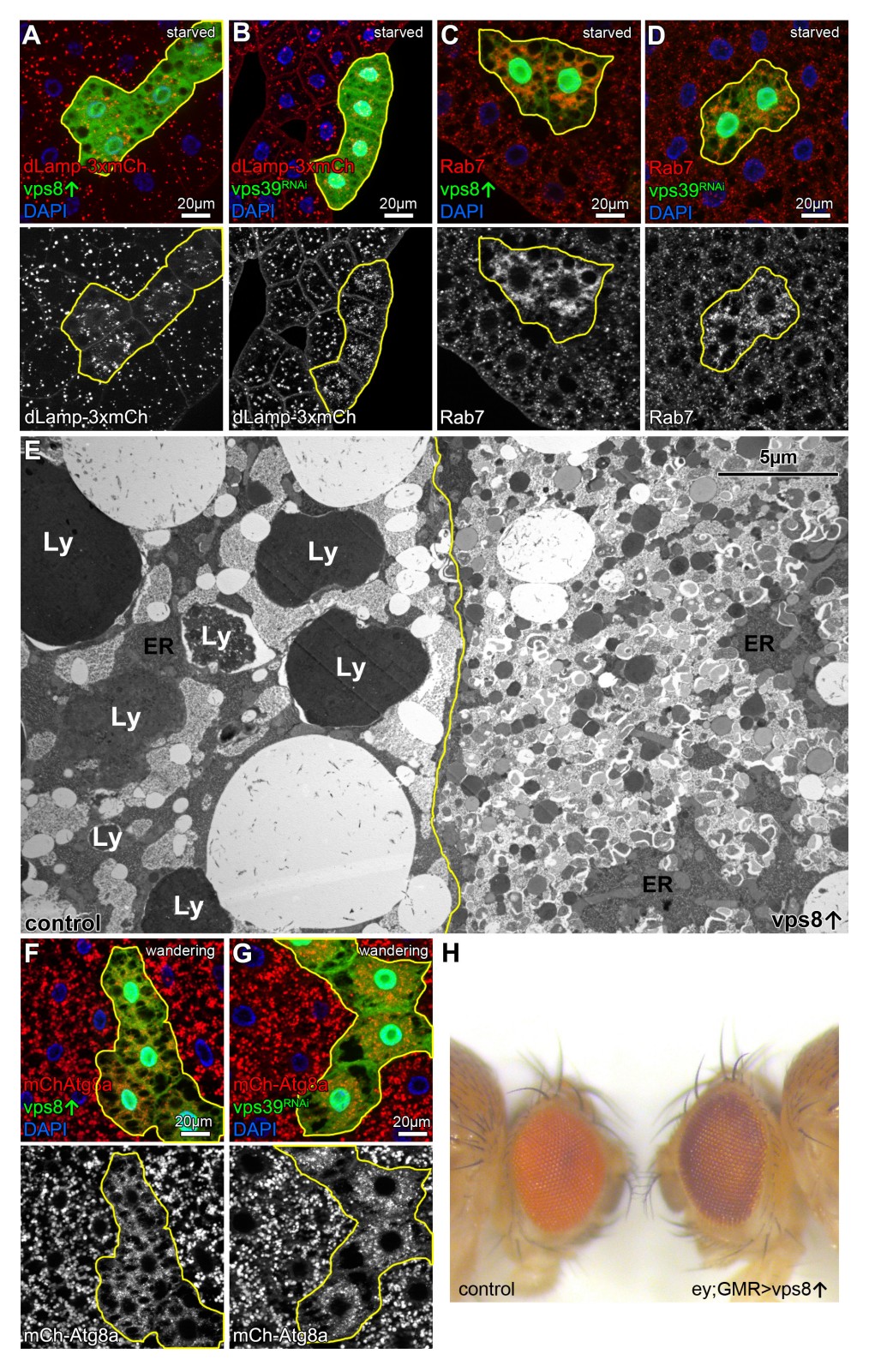

**Figure 4.** Overexpression of Vps8 impairs lysosome and LRO formation. (**A and B**) The size of lysosomes (marked by dLamp-3xmCherry) decreases, but their number increases in GFP marked starved fat cells overexpressing Vps8 (**A**) or *vps39* RNAi compared to neighboring GFP-negative control cells. (**C and D**) Starved, GFP+ fat cells that are overexpressing Vps8 (**C**) or *vps39* RNAi (**D**) are full of small Rab7+ vesicles, while the surrounding control cells contain fewer Rab7+ dots. (**E**) Developmental autophagy proceeds normally in a control fat cell (left) of a wandering staged larva, indicated by the
*Figure 4 continued on next page*

Figure 4 continued

presence of large lysosomes containing electron-dense, protein-rich material (Ly). In contrast, the Vps8 overexpressing cell (right) is devoid of such structures and is filled with double-membrane autophagosomes and small lysosomes. ER: endoplasmic reticulum. (F and G) Large 3xmCherry-Atg8a + autolysosomes are absent from GFP marked Vps8 overexpressing cells compared to neighboring control cells (F), similar to *vps39* RNAi cells (G) in the fat tissue of wandering larvae. (H) The bright red color of the compound eye of a control fly (left) becomes darker upon eye specific overexpression of Vps8 (right). Note that Vps8 overexpression also eliminates the pseudopupil, which is seen as a dark spot in the middle of the control eye (left).

DOI: https://doi.org/10.7554/eLife.45631.011

The following figure supplements are available for figure 4:

**Figure supplement 1.** Quantification of data from *Figure 4*, panels A–D, F and G.

DOI: https://doi.org/10.7554/eLife.45631.012

**Figure supplement 2.** Vps8 (miniCORVET) is dispensable for lysosome formation in starved fat cells.

DOI: https://doi.org/10.7554/eLife.45631.013

*Figure 5—figure supplement 1*). In contrast to its overexpression, *vps8* RNAi had no effect on the dLamp-3xmCherry or Rab7 pattern (*Figure 4—figure supplement 2*), in line with our model that the main roles of Vps8 may be restricted to certain cell types in flies (*Lőrincz et al., 2016a*).

Towards the end of the last larval stage, a rise of the molting hormone 20-hydroxyecdysone induces massive developmental autophagy to prepare larval tissues for elimination during/after metamorphosis (*Lőrincz et al., 2017a*; *Rusten et al., 2004*). The main ultrastructural characteristic of this stage in fat cells is the appearance of very large electron dense (auto)lysosomes, which are filled with intracellular components and larval serum proteins taken up from the blood (*Figure 4E*). In contrast, the ultrastructure of Vps8 overexpressing cells are remarkably different: no large lysosomes are present but numerous small lysosomes and autophagosomes fill the cytoplasm (*Figure 4E*). Accordingly, 3xmCherry-Atg8a also shows the absence of large autophagic structures: only small 3xmCherry dots fill the cytoplasm of Vps8 overexpressing or *vps39* RNAi cells (*Figure 4F,G*, *Figure 4—figure supplement 1*).

Taken together, these data suggest that overexpression of Vps8 inhibits autophagosome-lysosome fusion via the loss of HOPS, which also impaired both starvation-induced and developmental autophagy in flies (*Takáts et al., 2014*). As both yeast and mammalian HOPS complexes are required for autophagosome clearance (*McEwan et al., 2015*; *Rieder and Emr, 1997*), it will be interesting to test whether the overexpression of CORVET specific subunits also affects autophagy and related processes in these organisms.

## Overexpression of Vps8 inhibits HOPS dependent lysosome-related eye pigment granule biogenesis and crinophagy

Overexpression of Vps8 inhibited the two main HOPS dependent lysosomal degradation pathways, but the question arose whether elevated Vps8 levels can inhibit all processes in which HOPS is involved in. For example, HOPS is essential for lysosome related organelle biogenesis that includes the pigment granules of the fly compound eyes: several classical eye pigmentation mutants have been shown to carry mutations in genes encoding HOPS subunits (*Lloyd et al., 1998*; *Lőrincz et al., 2016b*). Thus, we overexpressed Vps8 in the eyes using combined ey-Gal4 and GMR-Gal4 drivers, which was previously shown to drive efficient expression of transgenes throughout eye development, including *vps16a* RNAi (*Pulipparacharuvil et al., 2005*). 5-day-old (after emerging from the pupal case) Vps8 overexpressing adults had darker eyes compared to age-matched controls, and the pseudopupil was completely absent: instead, the center of the eye appeared lighter (*Figure 4H*). This phenotype is remarkably similar to the phenotype of animals homozygous for the $lt^1$ classical eye color mutant (*Chyb and Gompel, 2013*) caused by a hypomorphic allele of the gene *light,* encoding the HOPS subunit Vps41 (*Warner et al., 1998*).

Crinophagy is an *atg* gene-independent special autophagic process in which excess secretory granules directly fuse with lysosomes in a HOPS-dependent manner (*Csizmadia et al., 2018*). Larval salivary glands are an excellent model to study crinophagy because at the beginning of metamorphosis, excess glue protein containing secretory granules are broken down this way. This can be monitored using combined Glue-GFP and Glue-dsRed expression in salivary gland cells (*Csizmadia et al., 2018*). Similar to mCherry-GFP-Atg8a autophagic flux reporters, Glue-GFP but not Glue-dsRed is quenched inside acidic crinosomes of control or *vps8* RNAi cells, but GFP signal

persists in Vps8 overexpressing cells (*Figure 5A,B*, *Figure 5—figure supplement 1A–C*), indicating the impairment of crinophagy. At the onset of crinophagy, the lysosomal membrane protein GFP-Lamp1 forms 'rings' around granules designated for degradation upon glue granule-lysosome fusion, but in Vps8 overexpressing cells no rings are formed: instead, GFP-Lamp1 shows punctate pattern as a result of a fusion defect between lysosomes and secretory granules (*Figure 5C,D*, *Figure 5—figure supplement 1D*). Thus, the Vps8 overexpression phenotype again is indistinguishable from HOPS loss-of function (*Csizmadia et al., 2018*).

## Vps41 association to a subset of late endosomes requires Rab2, Rab7, all other subunits of HOPS, and it is lost upon Vps8 overexpression

We next analyzed the precise localization of our fully functional Vps41-9xHA reporter. As expected, Vps41-9xHA clearly overlapped with endogenous Rab7 at the rim of late endosomes in garland nephrocytes (*Figure 6A*, see also *Figure 1—figure supplement 2C*). Interestingly, Vps41-9xHA localized only to a sub-population of Rab7 positive endosomes but not to lysosomes (although adjacent CathL and Vps41-9xHA positive organelles were often seen), and sometimes only a Vps41-9xHA patch could be detected on the late endosomes (*Figure 6A*, *Figure 6—figure supplement 1*). Thus, Vps41 defines a novel late endosomal compartment in nephrocytes: 598/1739 of the Rab7 endosomes (34.02%) were Vps41-9xHA positive based on the analysis of 52 cells from nine animals, suggesting that only mature endosomal membranes (ready to fuse with lysosomes) acquire Vps41/HOPS in order to avoid premature fusion.

We next asked the question how Vps8 overproduction affects Vps41/HOPS function. Vps41 association with a subset of late endosomes in nephrocytes (*Figure 6A,F*) is lost upon Vps8 overexpression: Vps41-9xHA becomes dispersed in the cytoplasm and accumulates in the nucleus in these cells (*Figure 6B,F*, *Figure 6—figure supplement 2D*).

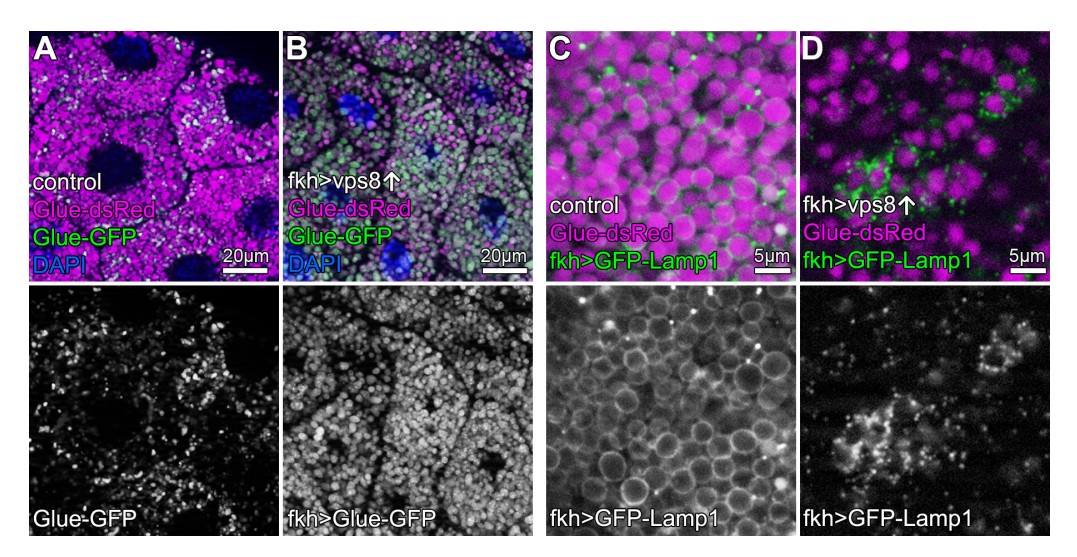

**Figure 5.** Overexpression of Vps8 impairs crinophagy. (**A** and **B**) Crinophagy in salivary glands at puparium formation. Glue granule degradation proceeds normally in control cells (**A**) based on the lysosomal quenching of Glue-GFP, with Glue-dsRed+, Glue-GFP- structures representing crinosomes. (**B**) Glue-GFP signal persists upon salivary gland-specific overexpression of Vps8, indicating an impairment of crinophagic flux. (**C,D**) Secretory granule-lysosome fusion is inhibited upon Vps8 overexpression. 2 hr before pupariation, lysosomes (marked by Lamp1-GFP) fuse with secretory granules as seen by the formation of Lamp1-GFP 'rings' at the rim of Glue-dsRed vesicles in controls (**C**). In contrast, Lamp1-GFP lysosomes accumulate and no Lamp1-GFP rings are present in Vps8 overexpressing cells (**D**), indicating impaired crinosome formation due to the lack of secretory granule-lysosome fusion.

DOI: https://doi.org/10.7554/eLife.45631.014

The following figure supplement is available for figure 5:

**Figure supplement 1.** Vps8 (miniCORVET) is dispensable for crinophagy.

DOI: https://doi.org/10.7554/eLife.45631.015

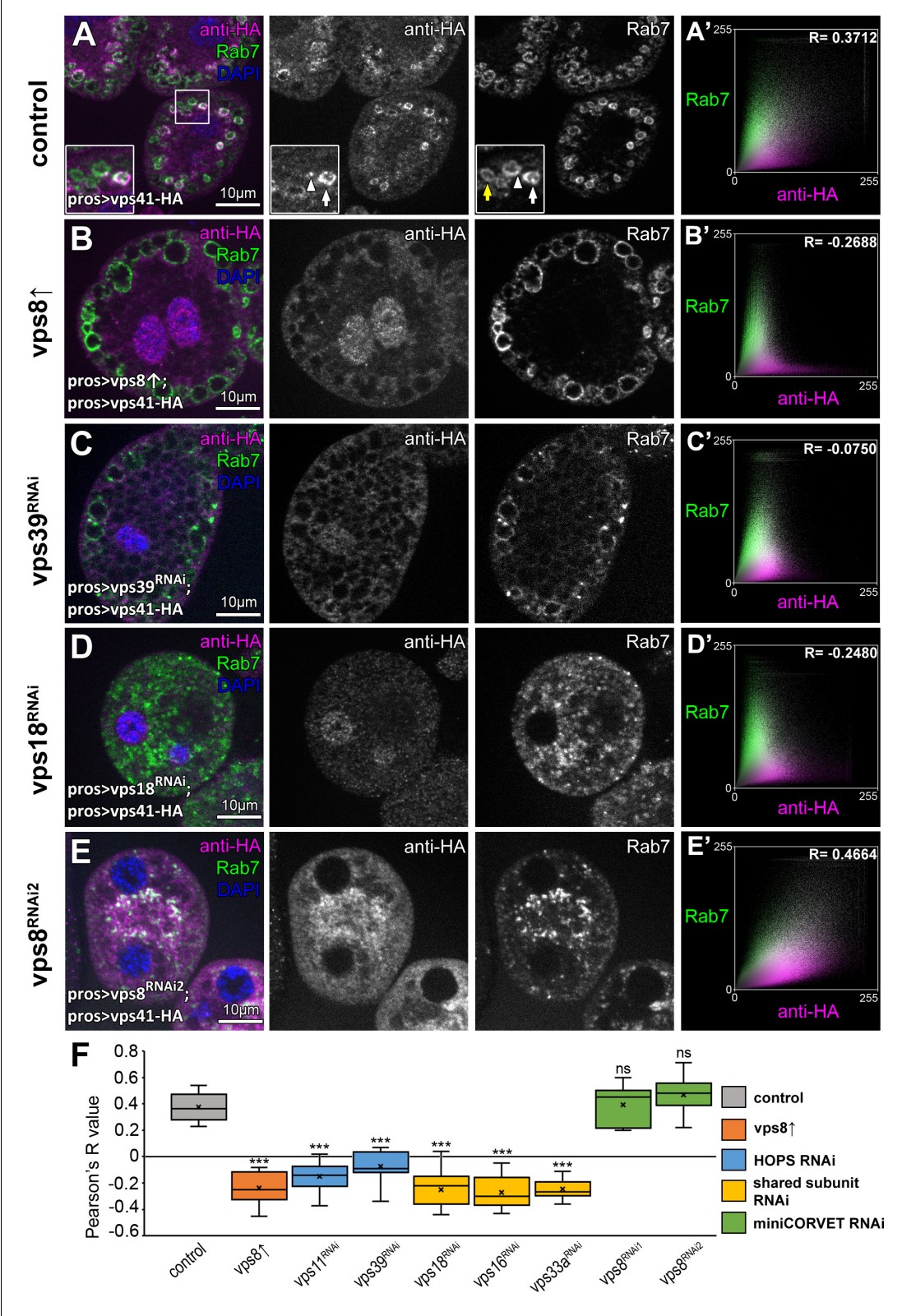

**Figure 6.** The late endosomal localization of Vps41 is lost upon Vps8 overexpression or loss of any other HOPS subunit. All images show nephrocytes expressing Vps41-9xHA in different genetic backgrounds, stained with anti-HA (magenta) and anti-Rab7 (green). (**A–E**) Vps41-9xHA is recruited to a subset of Rab7+ endosomes (white arrow) in control nephrocytes (**A**). White arrowhead points to a patch of Vps41-9xHA on a Rab7+ endosome. Yellow arrow points to a Rab7 endosome with no Vps41-9xHA signal. Upon Vps8 overexpression (**B**) or HOPS-specific *vps39* RNAi (**C**), nephrocytes become

*Figure 6 continued on next page*

*Figure 6 continued*

swollen and contain enlarged Rab7 endosomes. Strikingly, no Vps41-9xHA is detected on these vesicles. Moreover, no Vps41-9xHA can be detected on Rab7+ vesicles in *vps18* class C RNAi cells (D), and these late endosomes are fragmented due to the simultaneous loss of miniCORVET and HOPS. (E) Late endosomes are also fragmented and colocalize with Vps41-9xHA in *vps8* miniCORVET RNAi cells. (A'– E') Averaged scatter plots (generated from 18 control and 16 Vps8↑, 16 *vps39* RNAi, 15 *vps18* RNAi and 14 *vps8* RNAi cells) show the intensity correlation profile of Vps41-9xHA (labeled with anti-HA) with endogenous Rab7. Pearson correlation coefficients shown at the top of panels A' and E' indicate substantial colocalization, which is lost in Vps8 overexpressing, *vps39* and *vps18* RNAi cells (B'–D'). Quantification of colocalization data, including data from *Figure 6—figure supplement 1*. The median and the average of Pearson correlation coefficients are indicated as a horizontal black line and x within the boxes, respectively. Bars show the upper and lower quartiles, and significant differences are indicated in panels. ns: not significant difference, ***: p<0.001.

DOI: https://doi.org/10.7554/eLife.45631.016

The following figure supplements are available for figure 6:

**Figure supplement 1.** Vps41-9xHA is absent from Cathepsin L-containing lysosomes.
DOI: https://doi.org/10.7554/eLife.45631.017
**Figure supplement 2.** Additional Vps41-9xHA data.
DOI: https://doi.org/10.7554/eLife.45631.018

Importantly, the late endosomal localization of Vps41-9xHA is also lost upon HOPS-specific (*vps39* or *vps11*) or class C (*vps18*, *vps16A* or *vps33A*) RNAi (*Figure 6A,C–F*, *Figure 6—figure supplement 2B–D*), suggesting that Vps41 can associate to target membranes only as part of the fully assembled HOPS complex. Of note, Vps41-9xHA retains its late endosomal, Rab7-associated localization in *vps8* loss-of-function cells (*Figure 6E,F*, *Figure 6—figure supplement 2A*), indicating that miniCORVET has no role in promoting Vps41 recruitment to target membranes.

HOPS has been shown to be involved in the AP-3 pathway in yeast (*Angers and Merz, 2009*) and human Vps41 was suggested to regulate TGN to late endosome transport, possibly as an AP-3 coat in a HOPS independent manner (*Pols et al., 2013b*). Vps41-9xHA no longer associates to endosomes upon HOPS or class C loss and its distribution becomes mostly diffuse in these cells, suggesting that Vps41 functions predominantly as part of HOPS in Drosophila nephrocytes. In line with this, a recent study showed that both Vps41 and Vps39 are equally required for targeting lysosomal membrane proteins from the TGN to endosomes (*Lund et al., 2018*). However, we did not test the colocalization of Vps41 with AP-3 in nephrocytes or other tissues, so a HOPS-independent role of this protein cannot be excluded.

We have previously shown that Vps8 (miniCORVET) localization to early endosomes relies on Rab5 but not Rab7 (*Lőrincz et al., 2016a*), which motivated us to carry out functional analyses of selected small GTPases to identify additional factors required for Vps41/HOPS targeting. We have recently shown that Rab2 directly binds to the Vps39 end of HOPS, and Rab7 is also important for HOPS function even though it may bind this tethering complex only indirectly in Drosophila (*Lőrincz et al., 2017b*). In addition, lysosomal Arl8/Gie has also been identified as a HOPS binding small GTPase in Drosophila and human cells (*Khatter et al., 2015*; *Rosa-Ferreira et al., 2018*).

Indeed, we found that knockdown of either *rab7* or *rab2* leads to the cytoplasmic dispersion of Vps41-9xHA, and Vps41-9xHA association with vesicles positive for these Rabs in cells expressing their GTP bound (thus presumably constitutively active) YFP tagged mutant forms was increased compared to control cells (*Figure 7A–E*, *Figure 7—figure supplement 1D*). Interestingly, occasional Vps41 rings were still observed in *rab7* RNAi cells, in line with our model that it is Rab2 that directly binds HOPS and may be more important for its recruitment. Of note, ultrastructural analysis showed that *rab7* RNAi cells contain numerous enlarged endosomes and abnormal lysosomes (*Figure 7—figure supplement 1E*) just like HOPS depleted cells (*Figure 1H,I*), indicating that the mostly cytosolic localization of Vps41 is not due to the absence of endosomes, but it is a consequence of the absence of Rab7 itself.

Interestingly, Vps41-9xHA still retains its late endosomal association in Arl8/Gie depleted cells, although its overlap with Rab7 rings decreases (*Figure 7F*, *Figure 7—figure supplement 1D*). It is worth noting that in *rab5* depleted cells that are devoid of Rab7 positive vesicles, Vps41 is dispersed in the cytoplasm (*Figure 7—figure supplement 1A,D*), while cells expressing GTP-locked YFP-Rab5 contain numerous double Rab5 and Rab7 positive hybrid vesicles and Vps41-9xHA is never recruited to these (*Figure 7—figure supplement 1B,C*). This suggests that active Rab5 somehow prevents HOPS recruitment/assembly, perhaps via the maintenance of (mini)CORVET as previously we found

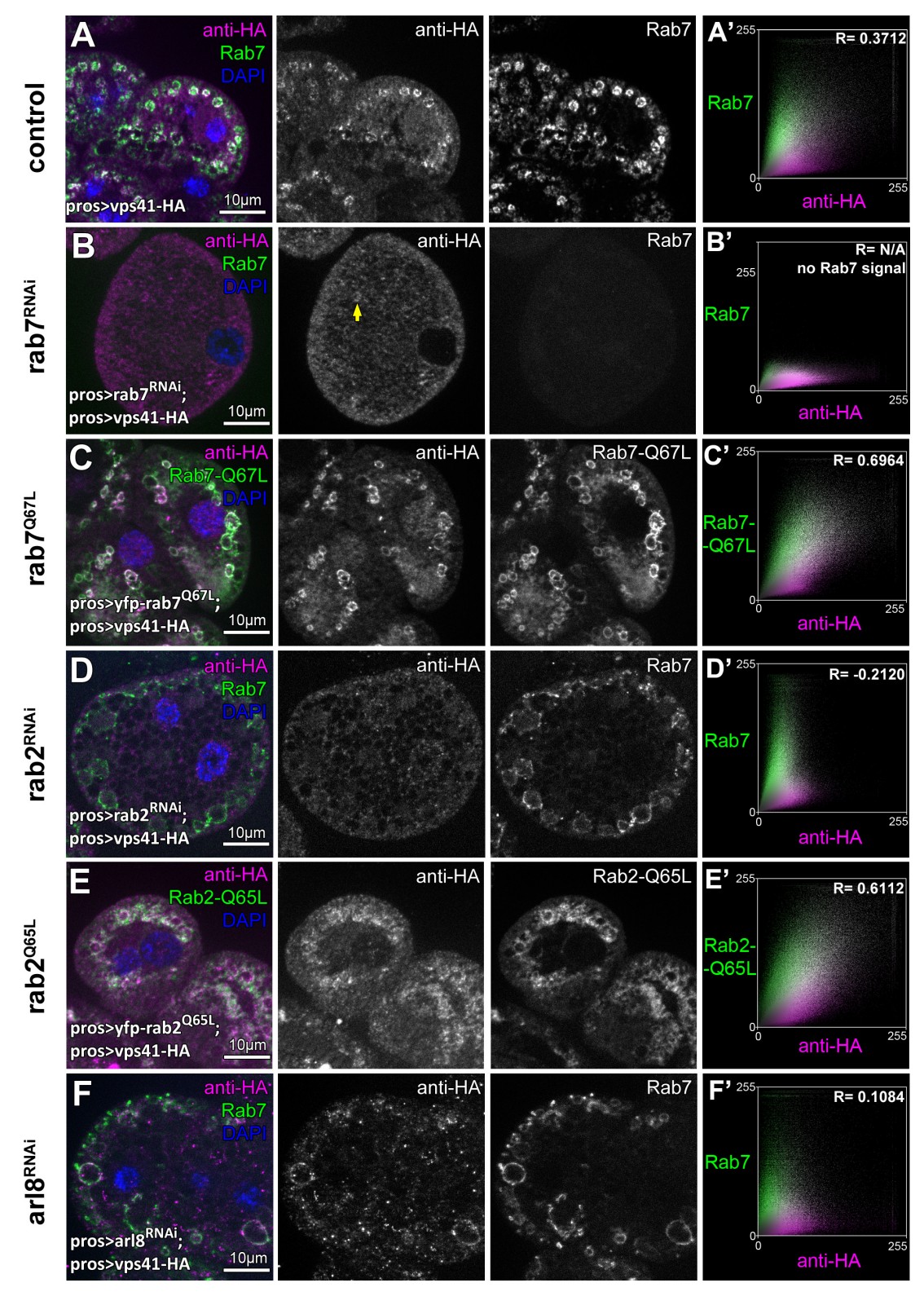

**Figure 7.** The late endosomal recruitment of Vps41 requires Rab2 and Rab7 but not Arl8. All images show nephrocytes expressing Vps41-9xHA on different genetic backgrounds, stained with anti-HA (magenta) and anti-Rab7 or YFP (green). (**A**) A subset of Rab7 endosomes are positive for Vps41-9xHA. (**B**) Vps41-9xHA is mostly dispersed throughout the cytoplasm in cells undergoing *rab7* RNAi. Interestingly, occasional Vps41-9xHA rings are still observed (arrow). (**C**) The colocalization of Vps41-9xHA with Rab7 increases on enlarged endosomes in cells expressing the constitutively active mutant

*Figure 7 continued on next page*

*Figure 7 continued*

form of Rab7. (D) Knockdown of *rab2* results in the swelling of nephrocytes that contain enlarged Rab7 endosomes with no Vps41-9xHA signal. (E) Vps41-9xHA is recruited to vesicles positive for GTP-locked Rab2. (F) The overlap of Vps41-9xHA with enlarged Rab7 vesicles decreases, but it is still obvious in cells undergoing *arl8* RNAi. Averaged scatter plots (generated from 13 *rab7* RNAi, 11 Rab-Q67L, 15 *rab2* RNAi, 11 Rab2-Q65L and 12 *arl8* RNAi cells) show the intensity correlation profiles of Vps41-9xHA (labeled with anti-HA) with Rab7 (B', D'. (F') or Rab7-Q67L (C') or Rab2-Q65L (E'). Pearson correlation coefficient was not determined in B' as no punctate Rab7 signal was detected. Pearson correlation coefficients shown at the top of panels A', B' and D' indicate substantial colocalization, no overlap in panel C', and still detectable colocalization in panel E'. Please note that since the experiments shown here and in *Figure 6* were carried out in parallel, the same averaged plot and Pearson correlation coefficient value is shown for controls (A') in both Figures.

DOI: https://doi.org/10.7554/eLife.45631.019

The following figure supplement is available for figure 7:

**Figure supplement 1.** Additional Vps41-9xHA data.

DOI: https://doi.org/10.7554/eLife.45631.020

that Vps8 is recruited to GTP-locked Rab5 positive vesicles, but not to GTP-locked Rab7 positive ones (*Lőrincz et al., 2016a*). These results together suggest that no shared compartments exist that would contain both miniCORVET and HOPS, also arguing against the model that these complexes are interconverted on target membranes in vivo.

## The amount of functional HOPS complex decreases upon Vps8 overproduction

Since the association of Vps41 and Vps39 to membranes is required for the stability of HOPS in yeast (*Cabrera et al., 2009*; *Ostrowicz et al., 2010*) and Vps18 recruits Vps41 to the human HOPS complex (*Hunter et al., 2017*), we reasoned that the amount of assembled HOPS may decrease upon Vps8 overproduction. To address this, we overexpressed Vps41-9xHA together with or without Vps8 systemically in larvae and looked at the amount of co-immunoprecipitated endogenous Vps16A, Vps18/Dor and Vps33A/Car class C proteins using Vps41-9xHA as bait. We found that in controls, Vps41-9xHA readily co-immunoprecipitated endogenous Vps16A, Vps18 and Vps33A, indicating that Vps41-9xHA is found in the Drosophila HOPS complex (*Figure 8*). Importantly, when Vps8 was co-expressed, the amount of co-precipitated endogenous class C proteins decreased to less than 9% of that observed in controls (*Figure 8*), showing that HOPS assembly is indeed impaired in these animals. Input samples contained equal amounts of proteins, so the overexpression of Vps8 did not decrease the expression of class C Vps core proteins.

## Final conclusions

We previously showed that the endosomal localization of Vps8 (and thus miniCORVET) depends on the class C proteins Vps18/Dor, Vps16A and Vps33A/Car (*Lőrincz et al., 2016a*). Similarly, we find that the endosomal localization of Vps41/Lt (and thus HOPS) requires all class C proteins and Vps39. Interestingly, miniCORVET is functional in the absence of HOPS specific subunits (Vps11, Vps39 and Vps41) in Drosophila and it localizes to early endosomes (*Lőrincz et al., 2016a*). Similarly, the eye pigmentation of *vps8* mutants is normal (*Lőrincz et al., 2016a*) and as shown here, the loss of Vps8 has no effect on autophagy and crinophagy based on both RNAi and mutant data. Moreover, Vps41 (HOPS) is still recruited to late endosomes in the absence of Vps8. These results indicate that HOPS is still functional in the absence of (mini)CORVET and vice versa. Consequently, the assembly and localization of (mini)CORVET and HOPS does not depend on each other (although the specific subunits may compete for the shared class C).

Currently two main models exist about class C Vps-based complex assembly: i. the dynamic subunit exchange model suggests that specific subunits of CORVET and HOPS sequentially dissociate from and associate with the Vps-C core, and ii. the independent complexes model suggests that CORVET and HOPS exist predominantly as discrete complexes that may or may not be stable. In the first model, dynamic remodeling of complex composition would accompany endosome maturation, while in the second model, separate preassembled complexes would associate and dissociate during transport (*Nickerson et al., 2009*). Dynamic subunit exchange models are in part based on yeast experiments in which hybrid/intermediate complexes could be isolated from cells overexpressing complex-specific subunits (*Ostrowicz et al., 2010*; *Peplowska et al., 2007*). However, the formation

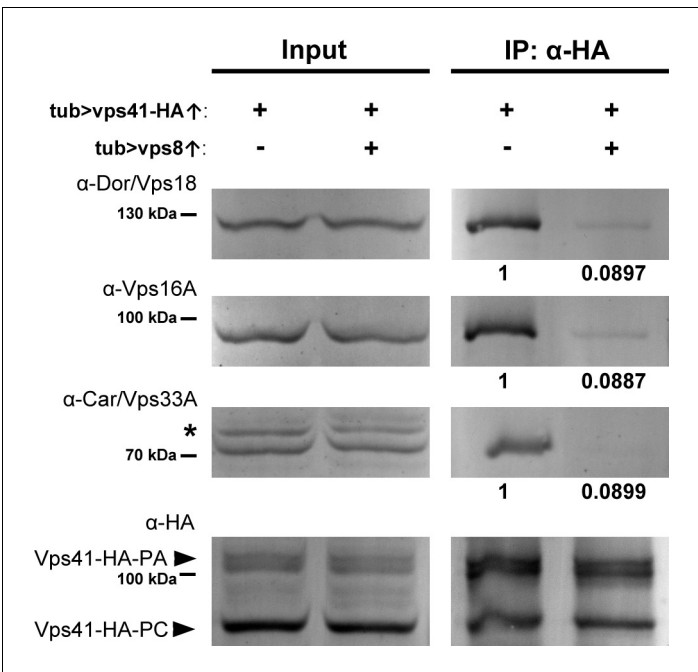

**Figure 8.** Vps8 overexpression strongly decreases Vps41 binding to the class C core. Endogenous Dor/Vps18, Vps16A and Car/Vps33A class C proteins coprecipitate with Vps41-9xHA isoforms (marked by PA and PC, respectively) from larval lysates. The amount of coprecipitated class C proteins decreases to less than 9% in lysates of larvae overexpressing Vps8. Asterisk marks a nonspecific band. Numbers under the bands in IP lanes refer to class C protein levels normalized to Vps41-9xHA PA+PC based on densitometry.
DOI: https://doi.org/10.7554/eLife.45631.021

of such complexes may be the result of the altered stoichiometric ratio of subunits. It is important to note that a Vps3-Vps41-class C hybrid complex was isolated from *vps8* or *vps39* mutant yeast cells (*Ostrowicz et al., 2010*; *Peplowska et al., 2007*), while *vps3* mutant cells contained an incomplete pentamer consisting of Vps8 and the four class C proteins but not Vps39 (*Ostrowicz et al., 2010*). This suggested that only three complexes (CORVET, HOPS, and the Vps3-Vps41-class C hybrid) are active as tethering or transition complexes in yeast (*Cabrera et al., 2013*). CORVET and HOPS was also suggested to assemble independent of each other, as lysates of *vps39* mutant cells contained a fully assembled CORVET complex (*Ostrowicz et al., 2010*).

Previously, we failed to detect hybrid complexes from fly lysates using a Vps8 bait expressed from its endogenous promoter (*Lőrincz et al., 2016a*), which may also be explained by the absence of a Vps3 homolog in Drosophila. In line with this, miniCORVET versus HOPS subunits were found in separate complexes purified using their corresponding small GTPase direct binding partners: Rab5 and Rab2, respectively. As neither Vps39 was detected in lysates of GTP-locked Rab5 expressing cells, nor Vps8 in lysates of GTP-locked Rab2 expressing cells, these data suggest that no chimeric complexes exist in fly cells (*Gillingham et al., 2014*). We thus favor the second model: the assembly of separate complexes independent of each other. Although we cannot exclude the possibility that intermediate complexes may form transiently and remain under the detection limit of co-IP and pull-down experiments, it is clear from our extensive loss-of-function experiments that (mini)CORVET is dispensable for autophagy, crinophagy and eye pigment granule biogenesis. Since HOPS is still functional in the absence of (mini)CORVET and vice versa, miniCORVET is functional in the absence of HOPS (*Lőrincz et al., 2016a*), these data also argue for the second model.

It was suggested that both CORVET and HOPS rely on the class C proteins as an assembly platform in yeast, because HOPS complex (and likely also the CORVET complex) disassembles in the absence of Vps11 (*Ostrowicz et al., 2010*). Interestingly, well-defined smaller subcomplexes could be isolated including a Vps39-Vps11 dimer from yeast cells lacking class C (except Vps11) proteins, suggesting that CORVET and HOPS assembly follows a specific order (*Ostrowicz et al., 2010*). We

found that in Drosophila, the localization of Vps41 depends not only on HOPS binding partners Rab2 and Rab7, but it also requires all class C Vps proteins, and the same is true for Vps8: its recruitment to early endosomes requires Rab5 and class C subunits (except Vps11). Thus, besides cytosolic assembly followed by recruitment, one possible variation of the second model is that class C-based tethering complexes are initiated by the binding of one end to a target membrane, followed by step-by-step assembly. This would mean that miniCORVET assembly is initiated by Vps18 binding to Rabenosyn-5, followed by sequential recruitment of Vps16A, Vps33A and lastly, Vps8, but Vps8 is still recruited by Rab5 in the absence of Rbsn-5 (*Lőrincz et al., 2016a*). Likewise, HOPS assembly might be initiated by Vps39 binding to Rab2 (or perhaps indirectly to Rab7, but we failed to detect Vps41 rings on Rab7-positive late endosomes in *rab2* mutants while occasional Vps41 rings could be observed in the absence of Rab7), followed by the sequential recruitment of Vps11, Vps18, Vps16A, Vps33A and finally Vps41. Future studies are necessary to decide between these alternative HOPS assembly scenarios.

In summary, we propose that Drosophila miniCORVET and HOPS assemble independent of each other, possibly in the cytosol, because:

1. The localization of Rab5-binding Vps8 requires the class C proteins Vps18, Vps16A and Vps33A (*Lőrincz et al., 2016a*), so miniCORVET is likely recruited to endosomes as a complex,
2. No HOPS-specific proteins were identified using Drosophila Vps8 as bait (*Lőrincz et al., 2016a*) and flies lack Vps3 so the Vps3-Vps41 hybrid class C complex cannot exist,
3. Vps41/Lt association with a subset of late endosomes is lost during HOPS loss-of-function caused by Vps8 overexpression or knockdown of other HOPS subunits (including the silencing of all class C genes and *vps39*), raising the possibility that pre-assembled HOPS is recruited to late endosomes instead of Vps41/Lt alone or smaller subcomplexes,
4. Vps8/miniCORVET is dispensable for autophagy, crinophagy and eye pigment granule biogenesis while HOPS is clearly required for these processes, so HOPS assembly does not depend on miniCORVET.

Assembled (mini)CORVET and HOPS tethering complexes are then recruited to target membranes by GTP bound Rab proteins. Importantly, CORVET-specific subunits better compete for the class C core against HOPS-specific proteins both in yeast (*Markgraf et al., 2009*; *Ostrowicz et al., 2010*; *Peplowska et al., 2007*) and flies (shown here). This could explain the relatively low and restricted expression of Vps8 in *Drosophila* (*Lőrincz et al., 2016a*), because higher or more widespread expression could interfere with essential HOPS functions.

# Materials and methods

**Key resources table**

| Reagent type (species) or resources | Designation | Source or reference | Identifiers | Additional information |
|---|---|---|---|---|
| Genetic reagent (*D. melanogaster*) | UAS-Vps8 | This study. | | |
| Genetic reagent (*D. melanogaster*) | UAS-Vps41-9xHA | This study. | | |
| Genetic reagent (*D. melanogaster*) | vps8[1]: vps8[1] | (*Lőrincz et al., 2016a*) | FBal0320420 | |
| Genetic reagent (*D. melanogaster*) | lt[ll]: lt[LL07138] | (*Lőrincz et al., 2016a*) | FBal0320422 | obtained from: DGGR |
| Genetic reagent (*D. melanogaster*) | vps11[LL]: vps11 [LL06553] | (*Takáts et al., 2014*) | FBal0296360 | obtained from: DGGR |
| Genetic reagent (*D. melanogaster*) | vps16a[d32] | (*Takáts et al., 2014*) | FBal0296357 | |
| Genetic reagent (*D. melanogaster*) | pros-Gal4 | | FBtp0129317 | obtained from: Bruce Edgar |
| Genetic reagent (*D. melanogaster*) | tub-Gal4 | | FBtp0020111 | obtained from: BDSC |

*Continued on next page*

*Continued*

| Reagent type (species) or resources | Designation | Source or reference | Identifiers | Additional information |
|---|---|---|---|---|
| Genetic reagent (*D. melanogaster*) | gen-vps8-HA | (*Lőrincz et al., 2016a*) | FBal0320419 | |
| Genetic reagent (*D. melanogaster*) | vps39$^{RNAi}$: vps39 [GD12152] | (*Lőrincz et al., 2016a*) | FBal0205346 | obtained from: VDRC |
| Genetic reagent (*D. melanogaster*) | vps11$^{RNAi}$: vps11 [KK102566] | (*Lőrincz et al., 2016a*) | FBal0231866 | obtained from: VDRC |
| Genetic reagent (*D. melanogaster*) | vps16a$^{RNAi}$: vps16 [GD13782] | (*Lőrincz et al., 2016a*) | FBal0208987 | obtained from: VDRC |
| Genetic reagent (*D. melanogaster*) | vps33a$^{RNAi}$: car [GD1397] | (*Lőrincz et al., 2016a*) | FBal0209225 | obtained from: VDRC |
| Genetic reagent (*D. melanogaster*) | vps18$^{RNAi}$: dor [KK102176] | (*Lőrincz et al., 2016a*) | FBal0231650 | obtained from: VDRC |
| Genetic reagent (*D. melanogaster*) | rab7$^{RNAi}$: rab7 [GD40337] | (*Lőrincz et al., 2016a*) | FBal0208211 | obtained from: VDRC |
| Genetic reagent (*D. melanogaster*) | vps8$^{RNAi1}$: vps8 [KK100319] | This study. | FBal0230675 | obtained from: VDRC |
| Genetic reagent (*D. melanogaster*) | vps8$^{RNAi2}$: vps8 [10144 R-1] | This study. | FBal0270342 | obtained from: NIG-Fly |
| Genetic reagent (*D. melanogaster*) | rab2$^{RNAi}$: rab2 [GD34767] | (*Lőrincz et al., 2017b*) | FBal0208203 | obtained from: VDRC |
| Genetic reagent (*D. melanogaster*) | arl8$^{RNAi}$: arl8 [7891 R-2] | (*Boda et al., 2019*) | FBal0275763 | obtained from: NIG-Fly |
| Genetic reagent (*D. melanogaster*) | rab5$^{RNAi}$: rab5 [JF03335] | (*Lőrincz et al., 2016a*) | FBal0241752 | obtained from: BDSC |
| Genetic reagent (*D. melanogaster*) | UAS-YFP-Rab5-Q88L | | FBal0215394 | obtained from: BDSC |
| Genetic reagent (*D. melanogaster*) | UAS-YFP-Rab7-Q67L | | FBal0215400 | obtained from: BDSC |
| Genetic reagent (*D. melanogaster*) | UAS-Rab2-Q65L | | FBal0215385 | obtained from: BDSC |
| Genetic reagent (*D. melanogaster*) | dLamp-3xmCherry | (*Hegedűs et al., 2016*) | FBal0325101 | |
| Genetic reagent (*D. melanogaster*) | 3xmCherry-Atg8a | (*Hegedűs et al., 2016*) | FBal0325100 | |
| Genetic reagent (*D. melanogaster*) | UAS-Lamp1-GFP | (*Pulipparacharuvil et al., 2005*) | FBal0221465 | obtained from: Helmut Krämer |
| Genetic reagent (*D. melanogaster*) | fkh-Gal4 | (*Csizmadia et al., 2018*) | FBtp0013253 | |
| Genetic reagent (*D. melanogaster*) | Sgs3-DsRed | (*Csizmadia et al., 2018*) | FBal0268258 | |
| Genetic reagent (*D. melanogaster*) | Sgs3-GFP | (*Csizmadia et al., 2018*) | FBal0119388 | |
| recombinant DNA reagent | EST AT14809 | | FBcl0024753 | obtained from: DGRC |
| recombinant DNA reagent | EST LD33620 | | FBcl0304050 | obtained from: DGRC |
| Antibody | mouse monoclonal anti-Rab7 | DSHB: Rab7 (*Riedel et al., 2016*) | RRID:AB_2722471 | IHC: 1:10 |
| Antibody | rabbit polyclonal anti-CathL | Abcam: ab58991 | RRID:AB_940826 | IHC: 1:100 |
| Antibody | rat polyclonal anti-Atg8a | (*Takáts et al., 2013*) | | IHC: 1:300 |

*Continued on next page*

*Continued*

| Reagent type (species) or resources | Designation | Source or reference | Identifiers | Additional information |
|---|---|---|---|---|
| Antibody | rabbit polyclonal anti-Atg8a | (*Takáts et al., 2013*) | | WB: 1:5000 |
| Antibody | rabbit polyclonal anti-p62/Ref(2)p | (*Pircs et al., 2012*) | RRID:AB_2569199 | IHC: 1:1000 WB: 1:4000 |
| Antibody | mouse mono clonalanti-tubulin | DSHB: AA4.3 | RRID:AB_579793 | WB: 1:2000 |
| Antibody | rabbit polyclonal anti-Car/Vps33A | (*Sevrioukov et al., 1999*) | RRID:AB_2569524 | WB: 1:1000 |
| Antibody | rabbit polyclonal anti-Vps16A | (*Pulipparacharuvil et al., 2005*) | RRID:AB_2569229 | WB: 1:2000 |
| Antibody | rabbit polyclonal anti-Dor/Vps18 | (*Pulipparacharuvil et al., 2005*) | RRID:AB_2569230 | WB: 1:1000 |
| Antibody | rat monoclonal anti-HA | Roche: 3F10 | RRID:AB_2314622 | IHC: 1:80 WB: 1:2000 |
| Antibody | rabbit polyclonal anti-Rab5 | Abcam: ab31261 | RRID:AB_882240 | IHC: 1:100 |
| Antibody | chicken polyclonal anti-GFP | Invitrogen: A10262 | RRID:AB_2534023 | IHC: 1:1500 |
| Antibody | rabbit polyclonal anti-HA | Sigma-Aldrich: H6908 | RRID:AB_260070 | IHC: 1:100 |
| Antibody | rat polyclonal anti-Rbsn-5 | (*Tanaka and Nakamura, 2008*) | RRID:AB_2569807 | IHC: 1:1000 |
| Antibody | mouse monoclonal anti-HA-Agarose | Sigma-Aldrich: A2095 | RRID:AB_257974 | |
| Chemical compound, drug | LysoTracker Red | ThermoFisher Scientific: L7528 | | 1:1000 |
| Software, algorithm | SPSS 17 | IBM | RRID:SCR_002865 | |
| Software, algorithm | ImageJ | ImageJ | RRID:SCR_003070 | |

Key resources table: references indicate the description or validation of the given reagent. DGGR: Kyoto Stock Center - Drosophila Genomics and Genetic Resources, Japan, BDSC: Bloomington Drosophila Stock Center: Indiana University Bloomington, USA, VDRC: Vienna Drosophila Resource Center, Austria, Nig-Fly: Fly Stocks of National Institute of Genetics, Japan, DGRC: Drosophila Genomics Resource Center, USA, DSHB: Developmental Studies Hybridoma Bank, USA.

## Fly work and treatments

Flies were raised at 25°C on regular food. The following mutant lines, deficiencies and transgenes were used: *vps8[1]; lt[LL07138], vps11[LL06553], vps16a[d32], pros-Gal4* (gift of Bruce Edgar, ZMBH Heidelberg, Germany), *gen-Vps8-HA, Df(3R)ED5339, Df(3L)ED211* (*Lőrincz et al., 2016a*). vps39$^{RNAi}$: *vps39 [GD12152]*, vps11$^{RNAi}$: *vps11 [KK102566]*, vps8$^{RNAi1}$: *vps8 [KK100319]*, vps16a$^{RNAi}$: *vps16 [GD13782]*, car/vps33a$^{RNAi}$: *car [GD1397]*, dor/vps18$^{RNAi}$: *dor [KK102176]*, rab7$^{RNAi}$: *rab7 [GD40337]* and rab2$^{RNAi}$: *rab2 [GD34767]* lines were obtained from Vienna Drosophila Resource Center (VDRC), vps8$^{RNAi2}$: *vps8 [10144 R-1]*, arl8$^{RNAi}$: *arl8 [7891 R-2]* were obtained from Fly Stock National Institute of Genetics, Japan (NIG-Fly). rab5$^{RNAi}$: *rab5 [JF03335]*, UAS-YFP-Rab5-Q88L, UAS-YFP-Rab7-Q67L, UAS-Rab2-Q65L were obtained from Bloomington Drosophila Stock Center (BDSC).

Vps39$^{RNAi}$,vps11$^{RNAi}$, vps16a$^{RNAi}$, car/vps33a$^{RNAi}$, dor/vps18$^{RNAi}$: rab7$^{RNAi}$ and rab5$^{RNAi}$ were validated previously by us (*Lőrincz et al., 2016a*). Similarly, Arl8 RNAi was validated previously by us (*Boda et al., 2019*).

We generated GFP marked Gal4-expressing fat cell clones using *hs-Flp; UAS-GFP; Act>CD2>Gal4* and *hs-Flp; dLamp-3xmCherry, UAS-GFP; Act>CD2>Gal4, UAS-Dcr2*, or *hs-Flp;*

*3xmCherry-Atg8a, UAS-GFP; Act>CD2>Gal4, UAS-Dcr2* (**Hegedűs et al., 2016**). Starvations were performed by floating 95 hr old larvae in 20% sucrose for 4 hr at RT. We estimated autophagic flux by the tandem mCherry-GFP-Atg8a reporter as previously described (**Mauvezin et al., 2014**; **Nagy et al., 2015**) and lysosomal fusions using *hs-Flp; 3xmCherry-Atg8a, UAS-GFP-Lamp1; Act>CD2>Gal4, UAS-Dcr2* us (**Boda et al., 2019**).

*GMR-Gal4* (FlyBase ID: FBst0001104) came from BDSC, and *ey-Gal4* was a gift of Viktor Billes (Department of Genetics, Eötvös Loránd University, Budapest, Hungary). Tub-Gal4 was used to drive ubiquitous expression of transgenes used in co-IP experiments and western blots from adult flies. Transgenes were expressed in the salivary glands using fkh-Gal4 combined with UAS-GFP-Lamp1 (**Pulipparacharuvil et al., 2005**), Sgs3-DsRed and Sgs3-GFP (Glue-dsRed and GFP) (**Csizmadia et al., 2018**). The experimental genotypes that we analyzed are shown in *Supplementary file 1*.

## Construction of UAS-Vps8 and UAS-Vps41-9xHA transgenic *D. melanogaster* lines

Vps8 was amplified from EST AT14809 (Drosophila Genomics Resource Center, DGRC) using primers tactatgcggccgcATGTCGGAGCTTAAGGCCCCGTCGCTG and ctcgaggtacCTATATAAATCGCCTGGGCGGTG and cloned into pUAST vector as a Notl-Acc65I fragment. Transgenic UAS-Vps8 flies were generated by random insertion using w[1118] embryos and standard procedures (BestGene).

Vps41 was amplified from the EST LD33620 (Drosophila Genomics Resource Center, DGRC) using primers: tgtacagcggccgcATGGCTAAAGCGTTGCCGCTC and tctagaggtaccTTTCCCCACGGTTAACTTCCAAA and cloned into pGen-9xHA vector (**Lőrincz et al., 2016a**) as a Notl-Acc65I fragment. Vps41-9xHA was amplified from this vector with the primers: aacagatctgcggccgcATGGCTAAAGCGTTGCCGCTC and aaagatcctctagaggtaccCTAAGCGTAATCTGGAAC and cloned into pACU Vector (Addgene, Plasmid #58373) using NEBuilder HiFi DNA Assembly Master Mix to the Notl-Acc65I site. Transgenic UAS-Vps41-9xHA flies were generated by using y⁻w⁻;+; attP2 (3L:68A4) embryos and standard procedures (BestGene).

## Co-immunoprecipitations and western blots

Western blot was performed using adult lysates as earlier (**Takáts et al., 2013**). Co-immunoprecipitations were performed as earlier, using anti-HA coupled to agarose beads (Sigma-Aldrich, A2095) (**Lőrincz et al., 2016a**). Beads were finally boiled in 25 µl Laemmli sample buffer and processed for western blot. The following antibodies were used for western blots: rabbit anti-Atg8a 1:5000 (**Takáts et al., 2013**) and rabbit anti p62/Ref(2)p 1:4000 (**Pircs et al., 2012**), mouse anti-tubulin 1:2000 (AA4.3-s, DSHB), rabbit anti-Car/Vps33A 1:1000 (**Sevrioukov et al., 1999**), rabbit anti-Vps16A 1:2,000, rabbit anti-Dor/Vps18 1:1,000, (**Pulipparacharuvil et al., 2005**), monoclonal rat anti-HA 1:2000 (Roche, 3F10). Anti-Car/Vps33A, anti-Vps16A, anti-Dor/Vps18 antibodies were gifts of Helmut Krämer (UT Southwestern Medical Center, USA). Secondary antibodies were alkaline phosphatase-conjugated anti-rabbit, anti-mouse and anti-rat (all 1:5,000; Millipore). Blots were developed by using NBT/BCIP colorimetric substrate solution (VWR).

## Immunohistochemistry

Immunofluorescence analyses of nephrocytes and fat bodies were performed as described (**Lőrincz et al., 2016a**; **Takáts et al., 2013**). The following antibodies were used: mouse anti-Rab7 (1:10, DSHB) (**Riedel et al., 2016**), rabbit anti-CathL (1:100, ab58991; Abcam), rat anti-Atg8a (1:300) (**Takáts et al., 2013**), rabbit anti p62/Ref(2)p 1:1000 (**Pircs et al., 2012**), rabbit anti-Rab5 (1:100, Abcam, ab31261), chicken anti-GFP (1:1500, Invitrogen: A10262), rabbit anti-HA 1:100 (Sigma-Aldrich: H6908), monoclonal rat anti-HA 1:80 (Roche, 3F10), rat anti-Rbsn-5 (1:1,000) (**Tanaka and Nakamura, 2008**). Alexa Fluor 568 goat anti-mouse, Alexa Fluor 568 goat anti-rat, Alexa Fluor 488 goat anti-rabbit, Alexa Fluor 488 goat anti-rat, Alexa Fluor 488 goat anti-chicken (all 1:1,000, Invitrogen).

## LysoTracker red staining of fat bodies, salivary gland dissection

Fat bodies of starved larvae were dissected in ice cold PBS and then incubated in LysoTracker Red (1:1000 in PBS, Thermo Fisher Scientific) for 2 min at RT. Samples were rinsed three times, mounted in 80% glycerol in PBS containing DAPI and photographed immediately.

Salivary glands were dissected in ice cold PBS, fixed for 5 min in 4% paraformaldehyde and mounted in 90% glycerol in PBS containing DAPI. Samples were photographed immediately.

All stainings (including immunohistochemistry), co-IP-s, western blots and ultrastructural analyses have been carried out at least twice (biological replicates), with similar results.

## Microscopy and statistics

Fluorescent images of *Drosophila* fat cells or garland nephrocytes were obtained at room temperature with an AxioImager.M2 microscope (Carl Zeiss) with an ApoTome2 grid confocal unit (Carl Zeiss) using Plan-Apochromat 40x/0.95 NA Air (Carl Zeiss) objective for fat cells, and Plan-Apochromat 63x/1.40 Oil (Carl Zeiss) objective for nephrocytes. Grid confocal images were captured using an Orca Flash 4.0 LT sCMOS camera (Hamamatsu), and Zeiss Efficient Navigation two software (Carl Zeiss). In order to enhance focus depths images from eight consecutive focal planes (section thickness: 0.24 μm for nephrocytes and 0.55 μm for fat cells) were projected onto one single image. Microscope and imaging settings were identical for all experiments of the same kind. Images were processed in Zeiss Efficient Navigation 2 (Carl Zeiss) and Photoshop CS4 or CS6 (Adobe) to produce final figures. Images of salivary glands were taken at RT using an AxioImager Z1 microscope (Zeiss) equipped with an Apotome1 grid confocal unit using AxioCam MRm camera and EC Plan-Neofluar 40×/0.75NA and AxioVision SE64 Rel. 4.9.1 (Zeiss) software. Images were processed in Photoshop CS3 Extended (Adobe). Compound eyes were photographed on a Lumar V12 stereomicroscope (Carl Zeiss) equipped with AxioCam ERc5s camera (Carl Zeiss).

Fluorescence structures from original, unmodified single focal plane images were quantified using ImageJ. The signal threshold for the relevant fluorescent channel was set by the same person when quantifying one type of experiment. For endosome or nephrocyte size measurements, images were imported in ImageJ, cells were randomly selected and the size of endosomes or cells were measured manually. For fat body experiments, a GFP-positive fat cell was randomly selected, and one of its immediate neighbor GFP-negative control cells was randomly selected for quantification. Dots over 30 pixel$^2$ (0.8 μm$^2$) size were counted as autolysosomes. Please note that fat cell clones are spontaneously and randomly generated independent of each other in mosaic animals. In all cases, only cells with their nuclei in the focal plane were selected to make sure that both perinuclear and peripheral regions are included in quantifications. The colocalization of Glue-Red with Glue-GFP or GFP-Lamp1 was manually quantified by the same skilled researcher. In other cases colocalization was determined using ImageJ's coloc2 plugin to calculate Pearson's coefficients using original grayscale images of the examined channels (1 = perfect colocalization, 0 = no/incidental colocalization, −1 = mutually exclusive localization) and the Manders coefficients plugin was used to generate scatterplots. Photoshop CS6 (Adobe) was used to project raw scatterplots onto one single, averaged image. The quantified data were evaluated using SPSS17 (IBM). T tests were used for comparing two and ANOVA for comparing multiple samples that all showed normal distribution, and U tests for comparing two and Kruskal-Wallis tests for comparing multiple samples that contained at least one variable showing non-Gaussian data distribution. Please see *Supplementary file 2* for more details of statistical analyses. For quantifying western blots, the RGB image of the scanned membrane was loaded in ImageJ, then converted to 8-bit grayscale. Bands were measured using Gel Analyze tool: Analyze>Gels>Select First Lane, Analyze>Gels>Select Next Lane, Analyze>Gels>Plot lanes, then magic wand tool was used to measure the densities of the plotted bands. The densities of class C bands were normalized to the sum of the densities of Vps41-9xHA PA and PC bands to get the levels of class C proteins relative to Vps41-9xHA. Then class C proteins levels from Vps8 overexpressing lysates were calculated relative to the control.

## Electron microscopy

Ultrastructural analyses of nephrocytes were performed as described (*Lőrincz et al., 2016a*). Fat bodies of wandering staged mosaic animals were adhered to a poly-L-lysine–coated glass slide in a drop of fixative (3.2% paraformaldehyde, 1% glutaraldehyde, 1% sucrose, and 0.028% CaCl$_2$ in 0.1

N sodium cacodylate, pH 7.4). GFP channel was photographed immediately using an AxioImager Z1 microscope (Zeiss), AxioCam MRm camera and EC Plan-Neofluar 10×/0.3NA and AxioVision SE64 Rel. 4.9.1 (Zeiss) software. Fat bodies were then embedded on the slide. Clones were identified on toluidine-blue stained semi-thin sections. Ultrathin 70 nm sections were stained in Reynold's lead citrate and viewed at 80kV operating voltage on a JEM-1011 transmission electron microscope (JEOL) equipped with a Morada digital camera (Olympus) using iTEM software (Olympus).

## Acknowledgements

We thank Sarolta Pálfia and Mónika Truszka for technical assistance and colleagues and stock centers mentioned in the Materials and methods section for supporting our work by providing reagents. This work was funded by the Hungarian Academy of Sciences (LP-2014/2 to GJ; PPD-222/2018 to PL, BO/00652/17 to ZS-V), the National Research, Development and Innovation Office of Hungary (GINOP-2.3.2-15-2016-00006 and −00032, K119842, KKP129797 to GJ and PD124594 to ZS-V) and the ÚNKP New National Excellence Program of the Ministry of Human Capacities of Hungary (ÚNKP-18–2-II-ELTE-32 to LK and ÚNKP-18–4-ELTE-409 to ZS-V). The funders had no role in study design, data collection and analysis, decision to publish, or preparation of the manuscript.

## Additional information

### Funding

| Funder | Grant reference number | Author |
|---|---|---|
| Magyar Tudományos Akadémia | LP-2014/2 | Gábor Juhász |
| Magyar Tudományos Akadémia | PPD-222/2018 | Péter Lőrincz |
| Magyar Tudományos Akadémia | BO/00652/17 | Zsófia Simon-Vecsei |
| National Research Development and Innovation Office | GINOP-2.3.2-15-2016-00006 | Gábor Juhász |
| National Research Development and Innovation Office | GINOP-2.3.2-15-2016-00032 | Gábor Juhász |
| National Research Development and Innovation Office | K119842 | Gábor Juhász |
| National Research Development and Innovation Office | KKP129797 | Gábor Juhász |
| National Research Development and Innovation Office | PD124594 | Zsófia Simon-Vecsei |
| Ministry of Human Capacities | ÚNKP-18-2-II-ELTE-32 | Lili Anna Kenéz |
| Ministry of Human Capacities | ÚNKP-18-4-ELTE-409 | Zsófia Simon-Vecsei |

The funders had no role in study design, data collection and interpretation, or the decision to submit the work for publication.

### Author contributions

Péter Lőrincz, Conceptualization, Data curation, Supervision, Validation, Investigation, Visualization, Methodology, Writing—original draft, Writing—review and editing; Lili Anna Kenéz, Formal analysis, Investigation, Visualization, Methodology, Writing—review and editing; Sarolta Tóth, Formal analysis, Investigation, Visualization; Viktória Kiss, Tamás Csizmadia, Formal analysis, Investigation; Ágnes Varga, Resources, Investigation; Zsófia Simon-Vecsei, Validation, Investigation; Gábor Juhász, Conceptualization, Resources, Supervision, Funding acquisition, Investigation, Writing—original draft, Project administration, Writing—review and editing

Author ORCIDs
Péter Lőrincz (ID) https://orcid.org/0000-0001-7374-667X
Zsófia Simon-Vecsei (ID) http://orcid.org/0000-0001-7909-4895
Gábor Juhász (ID) https://orcid.org/0000-0001-8548-8874

Decision letter and Author response
Decision letter https://doi.org/10.7554/eLife.45631.027
Author response https://doi.org/10.7554/eLife.45631.028

## Additional files

### Supplementary files

• Supplementary file 1. Genotype of animals used in this study.
DOI: https://doi.org/10.7554/eLife.45631.022

• Supplementary file 2. Additional table showing statistical tests, N and p-values.
DOI: https://doi.org/10.7554/eLife.45631.023

• Transparent reporting form
DOI: https://doi.org/10.7554/eLife.45631.024

### Data availability

All data generated or analysed during this study are included in the manuscript and supporting files.

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
