## [Decision Letter]

Thank you for submitting your article "Vps8 overexpression inhibits HOPS-dependent trafficking routes by outcompeting Vps41/Lt" for consideration by *eLife*. Your article has been reviewed by three peer reviewers, including Alexey J Merz as the guest Reviewing Editor and Reviewer #1, and the evaluation has been overseen by Vivek Malhotra as the Senior Editor. The following individual involved in review of your submission has also agreed to reveal his identity: Christian Ungermann (Reviewer #3).

The reviewers have discussed the reviews with one another and the Reviewing Editor has drafted this decision to help you prepare a revised submission.

This manuscript presents follow-on experiments from a nice 2016 study also published in *eLife*. Here, the authors explore the consequences of Vps8 and Vps41 overproduction in a variety of cell types in *Drosophila*. In short, Vps8 overproduction mimics Vps41 loss of function, while Vps41 overproduction has minimal phenotypes (though these phenotypes are somewhat less comprehensively explored). An additional finding is that, unlike in results reported for yeast, Vps8 does not seem to be required for multiple forms of autophagy.

Summary:

The experiments are generally technically sound and thorough.

The authors' core argument is that their studies support models in which HOPS and (mini)CORVET assemble in the cytoplasm rather than being interconverted on-the-fly on the surface of endolysosomal membranes as they mature. The reviewers agree that the latter model is only weakly supported by existing data, but were not persuaded that the current study stands as a definitive refutation of that model. A key difficulty is that we do not know whether these complexes are assembled once, and are then stable en bloc for the remainder of their working lives, or whether they are interconverted dynamically (whether in the cytoplasm or at a membrane surface). The interpretation of either model hinges on experiments addressing that question, and that no laboratory (including those of at least two reviewers) have so far found a definitive way to do the needed experiments.

Essential revisions:

1) The reviewers do not consider the models for HOPS and CORVET function as controversial as suggested by the authors. A more detailed discussion of the strengths and weaknesses of each model, and a more explicit statement of the open experimental uncertainties is needed. The authors refer to the Peplowska study, which provided some initial ideas on CORVET and HOPS dynamics. However, follow up work did not really find evidence for remodeling (Markgraf et al., 2009; Ostrowicz et al., 2010), and also the review of Balderhaar and Ungermann did not make this point. The review of Nickerson, Brett, and Merz, 2009, discusses both models as plausible.

2) The authors show nicely in Figure 1 that the α vacuoles expand upon Vps8 overexpression. It would be very helpful if the authors would quantify their effect relative to the HOPS depletion rather than Vps41 overexpression. In addition, they should show an EM image of the HOPS depletion in comparison. While all their data are consistent with a strongly impaired defect in HOPS function, it would be helpful for the reader to observe these phenotypes side by side.

3) Figure 6C: The authors should quantify their observation as the blot quality is rather poor. If possible, they should replace the blot by a more convincing and less dark blot.

---

## [Author Response]

Essential revisions:1) The reviewers do not consider the models for HOPS and CORVET function as controversial as suggested by the authors. A more detailed discussion of the strengths and weaknesses of each model, and a more explicit statement of the open experimental uncertainties is needed. The authors refer to the Peplowska study, which provided some initial ideas on CORVET and HOPS dynamics. However, follow up work did not really find evidence for remodeling (Markgraf et al., 2009; Ostrowicz et al., 2010), and also the review of Balderhaar and Ungermann did not make this point. The review of Nickerson, Brett, and Merz (2009) discusses both models as plausible.

Thank you for this comment. Accordingly, we have toned down our statements (including changing the Abstract) and a more detailed discussion was added to the main text, where we compare our findings with yeast data, discuss both models and cite Nickerson et al., 2009 as appropriate. While our data strongly prefer the cytosolic assembly and recruitment model over on-membrane (or cytosolic) transformation in part because HOPS functions independent of Vps8 in autophagy, crinophagy and LRO biogenesis (also, Vps8 expression is practically undetectable in these tissues), we also discuss the possibility that only one end of miniCORVET (Vps18) and HOPS (Vps39) is recruited to the target membrane first, followed by step-by-step assembly. Please see subsection “Final conclusions” for details.

2) The authors show nicely in Figure 1 that the α vacuoles expand upon Vps8 overexpression. It would be very helpful if the authors would quantify their effect relative to the HOPS depletion rather than Vps41 overexpression. In addition, they should show an EM image of the HOPS depletion in comparison. While all their data are consistent with a strongly impaired defect in HOPS function, it would be helpful for the reader to observe these phenotypes side by side.

Thank you for the suggestion to include HOPS loss of function nephrocyte data. Accordingly, we repeated the overexpression experiments now including *vps39* and *vps11* RNAi samples both at confocal microscopy (this time using anti-Rab5 to label early endosomes instead of Rbsn-5) and the ultrastructural levels (please see updated Figure 1). The quantifications of cell and endosome sizes of Vps8 overexpressing or HOPS depleted cells (Figure 1A-E) are shown in Figure 1—figure supplement 1F. The effect of Vps8 overexpression is practically identical to the effect of HOPS knockdowns.

Please note that original panels A-E from the previous version of Figure 1 have been moved to Figure 1—figure supplement 1, and panel G shows their quantification.

3) Figure 6C: The authors should quantify their observation as the blot quality is rather poor. If possible, they should replace the blot by a more convincing and less dark blot.

Thank you for the suggestion to improve this blot, which was indeed overexposed. We have collected new lysates and repeated the coIP, this time analyzing one more class C protein: Vps16a, in addition to Vps18 and Vps33a. Accordingly, we have replaced the old blots with new, properly exposed ones and quantified band intensity in IP lanes. These show that Vps8 overexpression decreases the amount of endogenous class C proteins precipitated by Vps41-9xHA to less than 9% of that observed in controls. Please note that original Figure 6 has been expanded and a new Figure 7 has been included to show additional epistasis tests as requested, so these new coIP western blots are now shown in Figure 8.